# True random number generation using the spin crossover in LaCoO$_3$

Kyung Seok Woo [1,2,3], Alan Zhang[1], Allison Arabelo[4], Timothy D. Brown [1], Minseong Park [1,2], A. Alec Talin [1], Elliot J. Fuller [1], Ravindra Singh Bisht[5], Xiaofeng Qian [4], Raymundo Arroyave[4], Shriram Ramanathan[5], Luke Thomas[6], R. Stanley Williams [1,2] ✉ & Suhas Kumar [1] ✉

While digital computers rely on software-generated pseudo-random number generators, hardware-based true random number generators (TRNGs), which employ the natural physics of the underlying hardware, provide true stochasticity, and power and area efficiency. Research into TRNGs has extensively relied on the unpredictability in phase transitions, but such phase transitions are difficult to control given their often abrupt and narrow parameter ranges (e.g., occurring in a small temperature window). Here we demonstrate a TRNG based on self-oscillations in LaCoO$_3$ that is electrically biased within its spin crossover regime. The LaCoO$_3$ TRNG passes all standard tests of true stochasticity and uses only half the number of components compared to prior TRNGs. Assisted by phase field modeling, we show how spin crossovers are fundamentally better in producing true stochasticity compared to traditional phase transitions. As a validation, by probabilistically solving the NP-hard max-cut problem in a memristor crossbar array using our TRNG as a source of the required stochasticity, we demonstrate solution quality exceeding that using software-generated randomness.

The increased prevalence of the Internet of Things (IoT) has led to large amounts of data being processed and exchanged[1,2]. This paradigm has necessitated both high-quality security and high-volume probabilistic computing. Both necessities require random number generation, which presently relies on pseudo-random number generators (PRNG) based on deterministic software algorithms being run on digital processors. This approach, due to its determinism, is vulnerable and is expensive in terms of the digital hardware needed to run the algorithms (such as the number of transistors). Put differently, highly precise digital hardware is combined with deterministic instructions to produce pseudo-stochastic information, which is less effective use of resources.

True random number generators (TRNGs), on the other hand, leverage unpredictable physical processes to generate truly random numbers. TRNGs enable both the trustworthiness of IoT ecosystems and high-speed probabilistic computing on large volumes of data. Research into TRNGs has attracted increased attention, with several switching mechanisms being employed for this purpose, such as Mott transitions[3], magnetic switching[4], etc. Memristors or memory resistors, constructed using such phase transition materials, due to their multiple degrees of freedom during the phase transitions (for instance, via coexisting phases), produce stochastic behavior and have been investigated as candidates for security applications[3,5-7]. Such physics-driven TRNGs are also inspired by the human brain's ability to generate stochasticity and chaos to accelerate probabilistic solutions to large data classification problems[8-11].

Here we demonstrate a TRNG using an electrical component (device) composed of LaCoO$_3$ (LCO) that undergoes a crossover

[1]Sandia National Laboratories, Livermore, CA, USA. [2]Department of Electrical and Computer Engineering, Texas A&M University, College Station, TX, USA. [3]Advanced Light Source, Lawrence Berkeley National Laboratory, Berkeley, CA, USA. [4]Department of Materials Science and Engineering, Texas A&M University, College Station, TX, USA. [5]Department of Electrical and Computer Engineering, Rutgers, The State University of New Jersey, Piscataway, NJ, USA. [6]Applied Materials Inc., Santa Clara, CA, USA. ✉e-mail: rstanleywilliams@tamu.edu; su1@alumni.stanford.edu

in the electron spin state, which results in a gradual insulator-to-metal transition (IMT). When electrically biased within the nonlinear current transport during the spin crossover, the component exhibits self-oscillations with a finite degree of stochasticity. This stochasticity is employed as an entropy source to generate random number sequences. We investigated the underlying causes of stochasticity through electrical measurements, analytical modeling, and phase field modeling. Our comprehensive approach revealed that the stochastic behavior, unlike in other phase transitions materials[12,13], is directly influenced by thermal fluctuations, which in turn introduce variations in material properties such as electrical conductivity. Our TRNG requires only a single circuit component, besides the LCO memristor, for binary bit generation and achieves the highest bit generation rate of 50 kb s$^{-1}$ among reported volatile-memristor-based TRNGs[3,5,6,14]. Furthermore, we demonstrate a nonvolatile-memristor-based Hopfield network using the LCO-based TRNG as a source of random fluctuations with a decaying noise profile to achieve simulated annealing. We show that such perturbations effectively escape local minima and find a global minimum for solving non-deterministic polynomial-time (NP)-hard problems in Hopfield networks. Our approach of using TRNGs as a true random number source outperforms software-equivalents that use a PRNG.

## Results

### Static behavior of LaCoO$_3$ memristor

Thin films of LCO were grown using pulsed laser deposition, with a thickness of 70 nm. Following film growth, we deposited two lithographically defined electrodes composed of 5 nm of Cr and 50 nm of Pt, with a component length of 5 μm (Fig. 1a). The quasistatic current-voltage (I-V) behavior of this component measured using a current sweep exhibits a region of current-controlled negative differential resistance (NDR), where the voltage reduces as current is increased (Fig. 1b). NDR is a signature of potential instability in an electro-thermal memristor, which can lead to dynamics such as oscillations[12,15,16]. In-situ x-ray absorption spectra obtained at different temperatures in the oxygen K-edge (Fig. 1c) confirm the known signatures of the spin crossover in our LCO film[17]. The O K-edge spectra around 530 eV are related to Co 3$d$ bands, and the peak at 529.5 eV shifted to a lower energy of 528.6 eV with higher temperature due to the spin-state transition from low ($t_{2g}^6$) to high ($t_{2g}^4 e_g^2$) spin state in Co$^{3+}$ ions. The gradual change in resistivity with temperature is also a signature of the spin-state transition (Fig. 1d)[18,19]. The spin crossover process has a more gradual change in the resistance compared to an abrupt change in a first-order phase transition (e.g., in Mott insulators[13]). NDR requires two conditions – first, increase in temperature upon increasing current (for thermally driven NDR); second, a minimum magnitude of non-linearity in the resistance decreasing as a function of temperature. Via in-situ thermal mapping at different current levels, we observed a relatively gradual temperature increase within the NDR region (Fig. 1e) in the order of ~20 K, satisfying the first criterion for NDR. Further, the three orders of magnitude decrease in resistance with increasing temperature (Fig. 1d), though gradual, provided sufficient nonlinearity to satisfy the second criterion required for NDR. Thus, the spin crossover is fundamentally responsible for the nature of the NDR and the dynamics associated with the NDR.

### Stochastic oscillations in LCO

When electrically biased in a region of NDR (using a current source), the LCO components exhibited self-sustained electrical oscillations in the form of repeated spikes (Fig. 2a and Supplementary Fig. 1). The

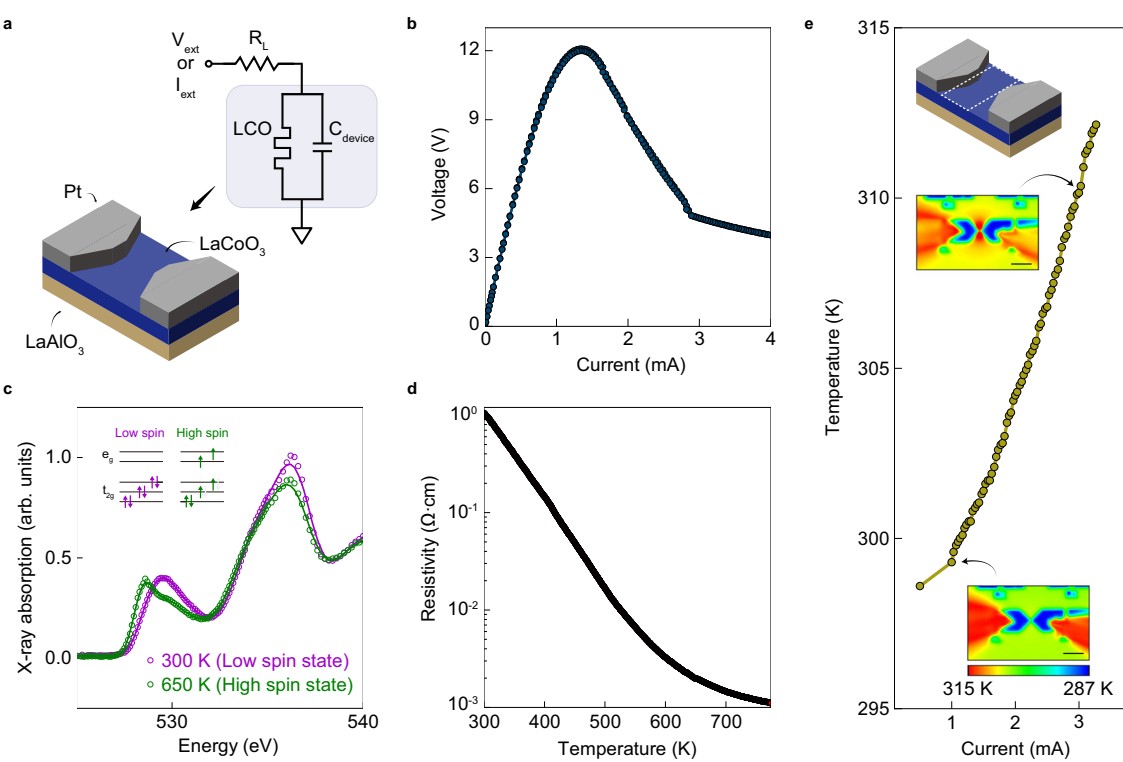

**Fig. 1 | LCO memristor. a** Schematic illustration of an oscillator circuit. The shaded region shows the memristor and its internal capacitor. **b** Quasistatic I-V curve with a current sweep mode. **c** X-ray absorption spectra (XAS) of LCO. The film was sputter-deposited specifically for X-ray measurements and was a different sample from the one used for the electrical measurements. Sputter deposition was required to enable growth on suspended silicon nitride membranes that allowed X-ray transmission at the oxygen K-edge. **d** Resistivity as a function of temperature. **e** In-situ thermal characterization of the LCO memristor at different current levels. The temperature of a dotted box region was measured (Top inset). The insets show thermal images at $I_{ext}$ values of 1 mA and 3 mA. The scale bar in the inset corresponds to 100 μm.

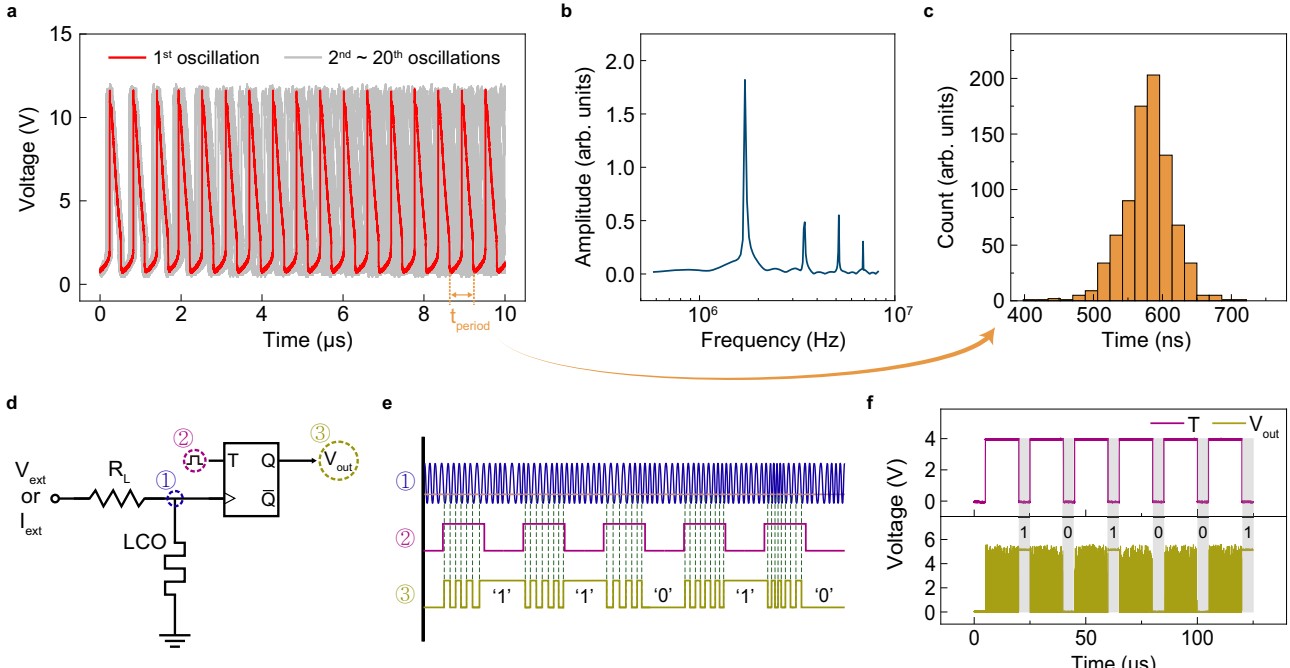

**Fig. 2 | LCO-based TRNG. a** Twenty sequential oscillations at $I_{ext}$ = 3.2 mA. **b** Fourier transform of the first oscillation in **a** to approximate the oscillation frequency (~2 MHz). **c** Distribution of time period ($t_{period}$) in oscillations. **d** Circuit model of the TRNG composed of a memristor and a flip-flop. **e** Working principle of LCO-based TRNG. **f** Experimental demonstration of six consecutive cycles producing random binary outputs.

load resistance was set to 2 kΩ. Such oscillations are attributed to the instabilities within a region of NDR and an additional degree of freedom in the form of an intrinsic capacitance (Fig. 1a)[16]. Since the oscillating time period (~0.5 μs) is roughly equal to the product of the load resistance (2 kΩ) and the internal capacitor, we estimate the internal capacitor to be a maximum of 0.3 nF. By comparing 20 different time series of oscillations (by aligning them to the first spike), we observed stochastic oscillating behavior, characterized by the absence of an overlapping oscillatory pattern (Fig. 2a, b). To statistically quantify the variations, we measured the time period of 800 oscillations from a single LCO component, revealing a substantial variation of roughly 25% (from the central time period) within a given component (Fig. 2c). We repeated this measurement on four different component, and all measured component exhibited similar stochastic variations, ensuring that the observed phenomena are not limited to a single component.

Using the stochastic oscillatory behavior of the LCO component, we constructed a prototype TRNG circuit by adding a negative-edge-triggered toggle (*T*) flip-flop (SN74LS73AN, Texas Instruments) (Fig. 2d and Supplementary Fig. 2). The working principle of our LCO-based TRNG is illustrated in Fig. 2e. The oscillator's output is directly applied as the clock signal to the flip-flop, while a periodic square-wave clock signal is applied to its toggle input. When *T* = 1 (high signal), the output flips (between 0 and 1) upon the negative edge of the clock signal. Due to the period stochasticity in the LCO oscillations, the output flipping and thus bit generation is random at every clock cycle. The experimental output of our TRNG passed the NIST randomness test[20] without any post-processing (Fig. 2f, Supplementary Table 1 and Supplementary Note 1). Notably, this TRNG outperforms previously reported volatile-memristor-based TRNGs with regard to bit generation rate, circuit simplicity, endurance, and energy consumption, as summarized in Table 1. Our work demonstrates the highest reported bit generation rate of 50 kb s⁻¹, which can potentially be enhanced to over 100 kb s⁻¹ (Supplementary Fig. 3), while only one flip-flop is required to build the TRNG. The LCO component was employed as the clock signal, which is energy efficient compared to other TRNGs that required an external clock generator. Kim et al. similarly leveraged the self-clocking ability

of a NbO$_2$ memristor[3]. Their approach, however, required an amplifier to increase the inherently low-current oscillating signal. Furthermore, our TRNG exhibits good endurance in that the LCO component oscillated over 12,000 seconds without any degradation, proving its capability to generate at least 600 M bits (Supplementary Fig. 4). The overall energy consumption of a TRNG primarily depends on the number of active components, with each component consuming milliwatts of power. The self-oscillation-based TRNGs offer energy advantages by eliminating the need for a clock generator (i.e., by reducing the number of peripheral components). A low-power clock generator (CDCI6214, Texas Instruments) consumes ~150 mW. Moreover, since the generated bit is based on the number of oscillations (bit flipping), the randomness of our TRNG can be tuned by adjusting the oscillating bias or *T* input pulse time (Supplementary Figs. 1 and 3). This tunable TRNG may present an efficient alternative to the time-consuming and energy-intensive process of rejection sampling used with PRNGs.

Memristors are increasingly employed as key components in TRNGs due to their inherent variabilities. In the early stages of memristor-based TRNG development, stochastic characteristics of nonvolatile memristors, such as current fluctuation, switching voltage variation, random telegraph noise, and delay/relaxation times were exploited[14,21–23]. However, these TRNG approaches face practical challenges, including circuit complexity, requirement of the RESET process, and reliance on post-processing steps, creating challenges for on-chip integration. To address these issues, there has been a shift in focus towards volatile-memristor-based TRNGs with self-OFF switching behavior, which can reduce energy consumption. Therefore, we compare the performance of volatile-memristor-based TRNGs that passed the NIST randomness test without post-processing (Table 1). The first volatile-memristor-based TRNG, which employed the stochastic delay time of an Ag:SiO$_2$-based diffusive memristor[5], successfully passed the NIST randomness tests without any post-processing, though it required a complex circuit with many components and produced a low bit generation rate. The present work, which expands the capabilities of volatile memristors by using a spin crossover

**Table 1 | Comparison of volatile-memristor-based TRNGs that passed NIST randomness test without post-processing**

|  | **This work** | **Jiang et al.**[5] | **Woo et al.**[6] | **Woo et al.**[14] | **Kim et al.**[3] |
|---|---|---|---|---|---|
| Component switching mechanism | Non-first-order phase transition | Diffusive | Electronic switching | Diffusive | First-order phase transition |
| Source of randomness | Oscillations | Delay time | Delay & relaxation times | Delay & relaxation times | Oscillations |
| Bit generation rate (kb s⁻¹) | 50 | 6 | 6 | 32 | 40 |
| TRNG circuit components (# of components) | T flip-flop only (1) | Comparator, AND gate, 2 T flip-flops (4) | 2 AND gates, T flip-flop (3) | XNOR gate, XOR gate, 4 D flip-flops (6) | Op-amp, T flip-flop (2) |
| TRNG endurance (# of bits produced per component) | 600 M | 54 M | Not reported (Two memristors scheme) | 48 M | 24 M |

material, expands the potential for highly reliable TRNGs that are compatible with post-digital hardware.

## Why is LCO better suited?

Our measurements suggest that crossover transitions could be inherently more effective than first-order phase transitions for building stochastic systems. Figure 1d revealed that the electrically-driven spin-state crossover in LCO leads to a more gradual transition relative to other materials, resulting in high endurance. Conversely, volatile switches driven by Mott transitions (e.g., in VO$_2$ and NbO$_2$) have a precipitous temperature-driven IMT, which can cause runaway switching events. Such abrupt variations lead to large local current densities and temperatures[24], which may result in material damage[25].

To understand the fundamental origin of the stochasticity in our components, we performed phase field modeling of LCO, based on first-principles calculations using material properties measured on our LCO films (Supplementary Note 2). The resulting free energy landscape (Fig. 3a) is strikingly different from a first-order phase transition. Firstly, either of the two spin states is likely to exist in a wide range of temperatures from 300 K to nearly 500 K. In most first-order phase transitions, a change from one phase to another occurs in a narrow window of temperature (or another control variable). Secondly, the spin gap between the two spin states at all temperatures up to 500 K is on the order of ambient thermal noise ~30 meV[26] (Fig. 3b). Such a low barrier essentially leads to a highly dynamical equilibrium between the two spin states. Though the system may obey global statistical distributions, there will be local volumes of LCO fluctuating between spin states due to ambient thermal fluctuations, which will likely affect other material properties as well. This possibility is confirmed in our calculation of the global high spin fraction (Fig. 3c) at various assumed levels of thermal fluctuations $\Delta$ (with $\Delta = k_B T$ representing ambient conditions, where $k_B$ is the Boltzmann constant). These global fractions were calculated as an average of many simulations of many instances with varying initial conditions and randomized fluctuations. For various levels of fluctuations, the high spin fraction is roughly 0.5 at room temperature (300 K). The various individual instances for two different cases are illustrated in Fig. 3d (for $\Delta = k_B T/10$ and $\Delta = k_B T$). For the case with lower assumed thermal fluctuation magnitude, nearly all the instances resulted in roughly the same high spin fraction at all temperatures. However, for ambient conditions, while the average of the high spin fraction was roughly 0.5 at room temperature (300 K), the individual instances exhibited a large variance. As expected, at low temperatures (less than 100 K), the system converged to either of the two spin states, trapped by the absence of appreciable thermal fluctuations. At high temperatures (above 600 K), the system tended towards the global average, driven by increased thermal fluctuations. At 300–500 K, there was a large variation, indicating not only coexisting spin fractions but also a high degree of sensitivity to thermal fluctuations. This large variation is the key factor that contributes to the stochastic oscillations even at room temperature. Furthermore, there is no sudden change in high spin fraction at any specific temperature, unlike first-order phase transition materials, which have

abrupt transitions causing structural damages during the switching[27]. In addition, Mott insulators that are routinely used to build oscillators undergo a transition at either very high temperatures (above 1000 K in the case of NbO$_2$[12,13]) or very low temperatures (about 340 K in VO$_2$[12,13]). Such transition temperatures are below the standard operating ambient temperature for commercial electronics (about 350 K) or very high (potentially damaging nearby materials if switching temperature is above 1000 K). LCO, on the other hand, has a transition in a broad range from room temperature up to about 700 K, which makes it suitable for chip operating environments. Therefore, LCO is a more stable on-chip material, as verified by our endurance testing and owing to its favorable transition temperature.

To experimentally quantify the existence of fluctuations, we measured the noise spectral density within low-bias currents at various ambient temperatures (Fig. 3e). The noise spectra exhibit an inverse frequency ($1/f$) dependence, which indicates that the current fluctuations likely drove a response that fed back into the system, such as temperature fluctuations that influenced conductivity. The noise spectral densities normalized to 10 Hz exhibit practically no variations across the temperature range of 285–355 K (Fig. 3f). While the observation of $1/f$ behavior in the raw noise spectra is an indication of thermal fluctuations driving an electrical quantity, such as conductivity, the absence of a temperature dependence is likely due to the activation energy for the physical processes responsible lying outside the temperature range investigated in this study. The stochastic behavior may be a manifestation of self-organized criticality[27]. As the system experiences thermally induced stochastic fluctuations, the system may self-organize into a critical state, contributing to the $1/f$ noise. The $1/f$ noise indicates that the spin crossover is not merely random but indicative of the system approaching a state of self-organized criticality.

We employed circuit-level Monte Carlo simulations to examine the effect of such fluctuations on the electrical dynamics of the component. We combined these simulations with a simplified compact model capable of exhibiting instability-driven oscillations[28–30]. We introduced fluctuations in various forms, including to the ambient temperature and to the thermal conductivity (Supplementary Note 3). These fluctuations resulted in oscillatory behavior that embodies stochasticity similar to the experimental observations (Fig. 3g, h and Supplementary Fig. 7). Thus, there is a clear connection between LCO's sensitivity to ambient fluctuations and its stochastic dynamics.

## Using TRNGs to solve optimization problems

After constructing a TRNG and identifying the underlying physics, we sought to demonstrate its practical utility and compare it to prevailing software-generated random numbers. For this demonstration, we chose to solve optimization problems, which are crucial in various applications. For instance, the maximum-cut (max-cut) graph partitioning problem, where the nodes of a graph are partitioned into two disjoint subsets to maximize the number of edges between them (Fig. 4a), is used in genome sequencing and efficient routing of signal paths in electronic circuits. The max-cut problem represents

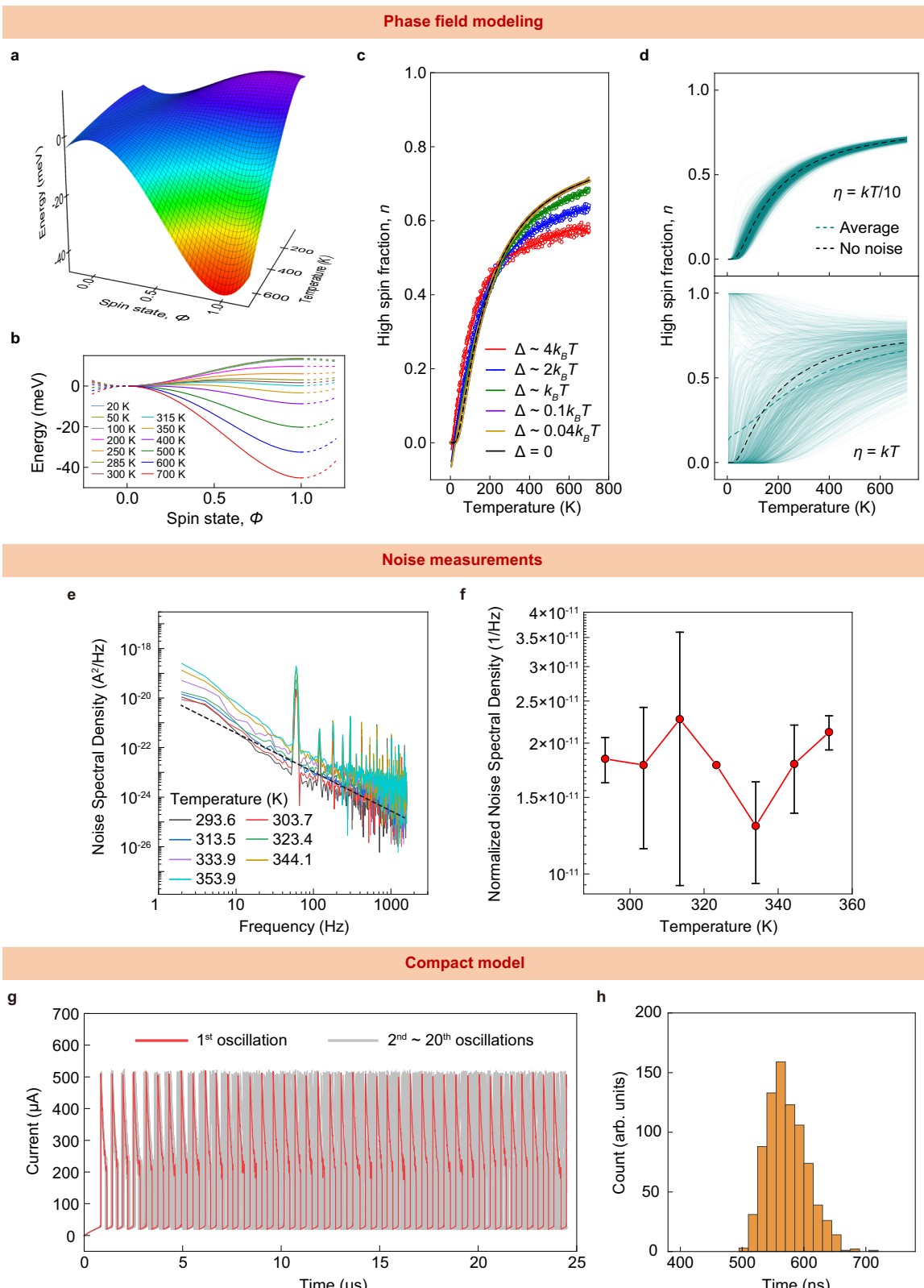

**Fig. 3 | Origin of stochastic spin crossover. a** Free energy landscape of LCO. **b** Free energy as a function of spin state at different temperatures. **c** High spin fraction as a function of temperature at different magnitudes of thermal fluctuation. The magnitude of the noise is set to be $\Delta(T) = k_B T$. **d** High spin fraction as a function of temperature at two different magnitudes of thermal fluctuation with constant noise ($\Delta = k_B \times 300$ K). **e** Noise spectral density of LCO at different temperatures with a 1/f slope (dashed line). The peak at 60 Hz is likely due to electrical interference or noise. **f** Noise spectral density, normalized to the spectral weight at 10 Hz at different temperatures. **g** 20 different simulated oscillations. **h** Distribution of time period in the simulated oscillations.

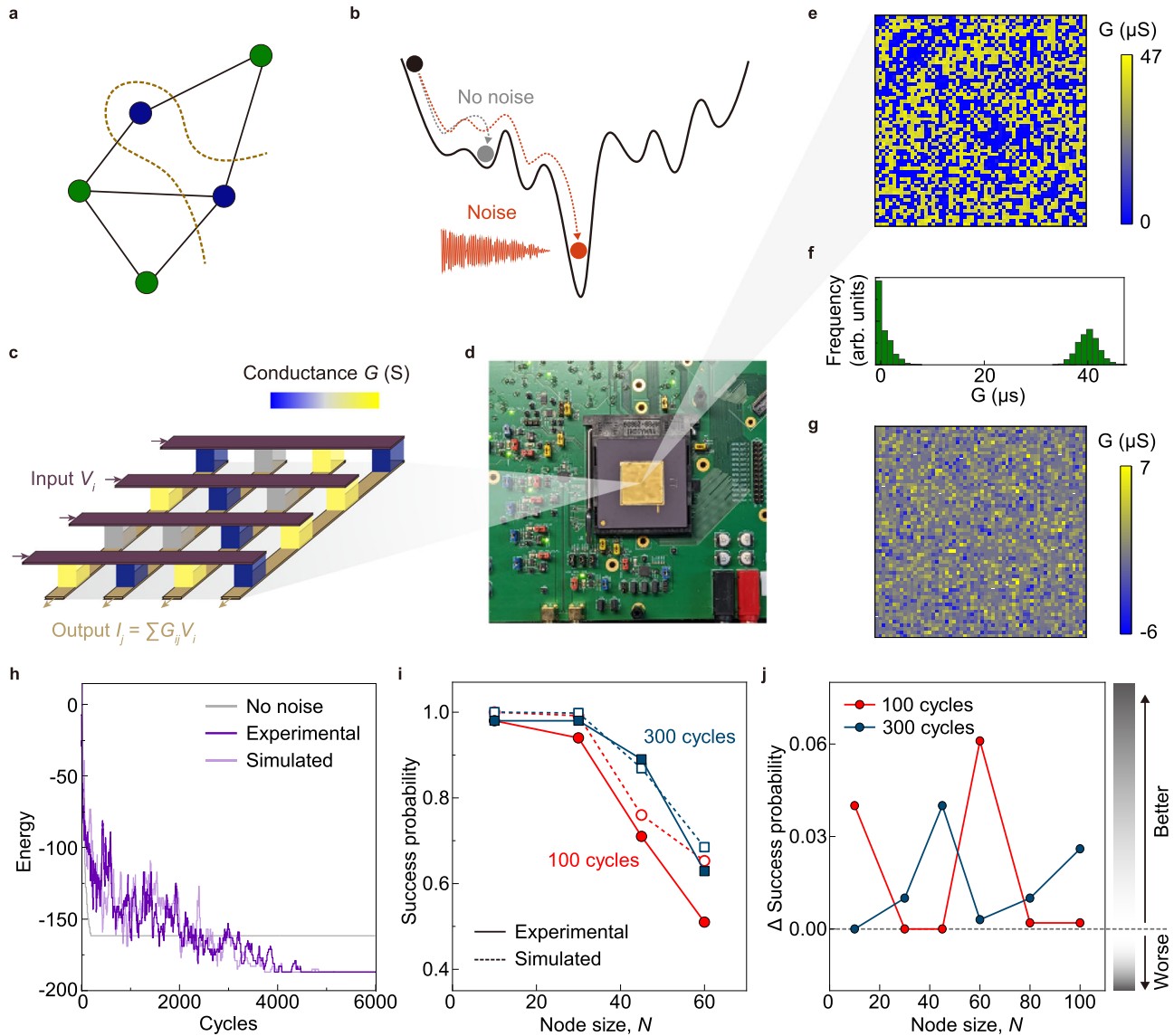

**Fig. 4 | Memristor-based noise-aided Hopfield network. a** Illustration of a max-cut NP-hard problem. **b** Energy landscape of a Hopfield network with and without noise. **c** Schematic of the memristor crossbar within the chip. **d** The chip used for the Hopfield network demonstration. **e** Experimental conductance-weight matrix for a problem of size $N = 60$, and **f** the corresponding conductance distribution. The conductance matrix represents the max-cut problem being solved. The relationship between the problem's graph and the conductance matrix is provided elsewhere[31]. **g** Normalized experimental error in the conductance matrix relative to the target (experimentally programmed conductance matrix minus the target conductance matrix). **h** Energy descent of 100 cycles for TRNG-based Hopfield network in calculations with no noise, hardware-realistic simulations (with hardware-matched noise), and experimental hardware results. Clearly, the case with no noise settles into a high-energy incorrect solution quickly and stays there, whereas the cases with realistic noise settle into a lower energy (optimal) solution. **i** Success probabilities of TRNG-based Hopfield network for 100 and 300 cycles at different node sizes. **j** Success probability of TRNG-based network minus that of PRNG-based network at different node sizes. Data points above zero on the vertical axis indicate superior performance compared to PRNGs.

generalized optimization and constrained optimization problems since it has full generality in terms of its representative Hamiltonian formulation[31]. Thus, our ability to improve solutions to the max-cut problem is a demonstration of improving solutions to any optimization problem. These problems cannot be efficiently solved using prevailing digital graphics processing units (GPUs) and central processing units (CPUs), owing to the complexity and the NP-hard nature of most such problems[32]. As such, probabilistic solutions to optimization problems are a practically viable option. Energy-based recurrent neural networks, such as Boltzmann machines[33,34] and Hopfield networks[35,36], have shown the potential to outperform conventional computers in probabilistic optimization. Most optimization problems are non-trivial, containing many local minima in their energy landscape, corresponding to sub-optimal solutions (Fig. 4b). The global energy

minimum of their energy landscape is the most optimum solution. Hopfield networks are known to get trapped in local minima during an energy minimization process, which presents a limitation to its efficacy in problem-solving. Noise is useful to help the network escape local minima through local energy ascent and potentially find the global minimum. Here we demonstrate a memristor-based Hopfield network using an LCO-based TRNG as a source of noise, where the noise was applied in a decaying fashion to implement simulated annealing.

The memristor-based Hopfield network was implemented using a crossbar array of oxide memristors designed for vector-matrix multiplication (Fig. 4c–g). Using such memristor crossbars to accelerate optimization with a Hopfield network has been discussed in detail[32], and the chip and supporting hardware are detailed elsewhere[37,38]. The noise was added to the system by using the outputs of the LCO-based

TRNG (stored in a separate memory unit and software-weighed). We adopted a decaying noise profile for better solution quality, which implemented simulated annealing[39–41]. The results agree with simulations of a circuit-accurate model of the system (with the experimental results slightly exceeding the simulations in quality), meaning optimal performance (Fig. 4h, i). The slightly differing performance exhibited by the experiments are likely due to the additional noise originating from the various components of the circuit (memristor conductance fluctuations, read circuit noise, etc.). Without any noise, the Hopfield network converged to a local minimum after a few cycles and could not escape from this state. Therefore, noise is indispensable in solving NP-hard problems that have complex energy landscapes. Such a memristor-based solution, when operating optimally, has previously been shown to outperform prevailing GPUs by over 5 orders of magnitude when scaled to sub-15 nm CMOS nodes via standard foundry rules[32,42].

Beyond showing that the TRNG can produce optimal performance in an experimental memristor-based Hopfield network, we sought to compare the TRNG's performance to that of a software-generated pseudo-random number generator (PRNG). A comparison (obtained using our circuit-accurate simulator) reveals a modest but measurable improvement in solution quality when a TRNG is used (Fig. 4j, Supplementary Figs. 8 and 9). This result may be ascribed to the fact that PRNGs are based on deterministic, though difficult to crack, algorithms. Such deterministic processes may be correlated to the dynamics of the Hopfield network, which diminishes their ability to detrap the system from local minima. In other words, the process used to disturb and dislodge the system must be as uncorrelated from the system's natural dynamics as possible, else, the dynamics and the dislodging process together will get stuck in newly resulting local minima. The TRNG outperforming PRNGs by 0.2–5% is an indirect but clear indication of this phenomenon that can be measured via Hopfield dynamics. The fundamental distinction between deterministic PRNGs and stochastic TRNGs (in the quality of the random bit streams) highlights that TRNGs have superior performance in probabilistic computing. The speed of our TRNG (sub-MHz range) is far lower compared to prevailing CMOS technologies (up to GHz range). This difference is attributed mainly to the micrometer-scale sizes of our laboratory-scale components compared to the CMOS technologies often manufactured at sub-10-nm sizes. As such, we expect the speed to increase notably upon scaling down the sizes of our prototype components and not pose a fundamental bottleneck. Combining a feedback shift register or utilizing a nanoscale heater could further increase the bit generation rate[28,43].

## Discussion

There are several more reported random number generators, which have been shown to pass one (or some) of the NIST tests, but not all of them. In Table 1, we included only those reports that demonstrated passing of all the NIST tests, because, as shown in prior works, failing one of the tests (e.g., the frequency monobit test) may lead to failures in several other tests[3,20]. Similarly, a full NIST test of processing at least 55 sequences is required to obtain statistically significant data. Further complicating a fair and quantitative comparison, different reported components were fabricated at different sizes and operated under different conditions, while many of them use discrete peripheral components assembled on breadboards (such as amplifiers)[3]. The performance metrics for some of them are reported as projections to cutting-edge technology nodes, such as a 7 nm node[44]. A fair comparison would require experimental demonstrations at identical technology nodes for both the component and its peripheral circuits. At the least, a comparison would need standardized design kits that enable simulated projections at a common technology node. As such, the state of the literature on TRNGs (and post-CMOS computing in general) is too nascent to engage in rigorous and quantitative comparisons, which will require more work on various types of TRNGs.

Despite the challenges in fairly and quantitatively comparing emerging TRNGs, here we provide a qualitative but useful comparison, which will aid the selection of the appropriate TRNG for a given application. We base our analysis on the fundamental limits of the underlying physical process used to generate random numbers and assume that the reported physical processes can lead to true randomness (by passing all the NIST tests). We broadly see electronic phase transitions and magnetic switching emerging as two promising processes for TRNGs. Pure electronic phase transitions that do not involve the movement of ions or significant changes in the crystal structure (similar to the spin transition in LCO or a Mott transition in $VO_2$) are likely the fastest in terms of fundamental speed limits (well below 1 ns)[45]. Magnetic tunnel junctions (MTJs) based on magnetic actuation likely follow with a timescale in the order of 1 ns[46]. Diffusive memristors, or those that rely on ionic motions, typically exhibit slower speeds of microseconds or more[5,14]. With regard to switching energy, superparamagnetic switching likely offers the lowest operating energies (in the order of 1 fJ per bit), but suffers from slower speeds[47]. We expect MTJs based on magnetic actuation and diffusive memristors to exhibit operating energies below 1 pJ per bit[5,46]. Electronic trapping/de-trapping switching mechanism also offers low energy consumption with high reliability[6,48]. Electronic phase transitions typically require thermal actuation in addition to the electric field driving Joule heating, resulting in higher energy consumption[3]. Therefore, there is no clear winner in terms of all the metrics of interest, but studies like ours enable the choice of an appropriate TRNG for a given application.

In summary, we experimentally demonstrated a memristor-based TRNG that exploits the inherent stochastic behavior of the spin crossover in $LaCoO_3$, while requiring only a single additional circuit component. Our compact and first principles models showed that the spin crossover is highly susceptible to thermal fluctuations, which results in stochastic oscillations. This compact TRNG not only sets a new standard with its superior bit generation rate but also demonstrates versatile applicability. Specifically, we used the output from this TRNG in a Hopfield network, harnessing its noise to assist the network in escaping local minima and thereby improving its accuracy. Electrical conductivity modulation resulting from spin fluctuations therefore opens a new direction for the discovery and design of semiconductors for probabilistic computing and cryptography.

## Methods

Device fabrication: An epitaxial thin film of LCO was grown in a Neocera pulsed laser deposition system (PLD) on a $LaAlO_3$ substrate. $LaCoO_3$ target was purchased from Toshima Manufacturing Co., Ltd. The substrate was etched in dilute HCl and annealed in air at 950 °C for 2 h. During the growth, the substrate temperature was 650 °C with an $O_2$ partial pressure of 100 mTorr. The PLD chamber pressure was increased to 2.5 Torr during cooldown. For Electrical measurements: The DC current-voltage (I-V) characteristics of the devices were measured using a Keysight B2911A Source Measure Unit. Self-oscillations in the NDR region were recorded using an Agilent Technologies MSO7054A oscilloscope.

NIST randomness test: NIST Statistical Test Suite (Special Publication 800-22) was run in Python, and 80 sequences of 1 M bits were collected for the test. Each test was considered passed if the P-value was higher than 0.001.

## Data availability

Due to the large size of data presented in the manuscript, instead of uploading the data along with the manuscript, the relevant data will be supplied by the corresponding authors upon request.

## Code availability

The codes used for phase field modeling are available at this URL: https://github.com/aiarabelo/LaCoO3_Thermodynamics/blob/main/Thermodynamic_Model_for_LaCoO3_(LCO).ipynb. The code can be run on Python, an open access tool. If needed, additional background information on the codes and support in running it can be obtained from the corresponding authors.

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

## Acknowledgements

This work was primarily supported as part of the Center for Reconfigurable Electronic Materials Inspired by Nonlinear Neuron Dynamics (reMIND), an Energy Frontier Research Center funded by the US Department of Energy (DOE), Office of Science, Basic Energy Sciences. This paper describes objective technical results and analyses. Any subjective views or opinions that might be expressed in the paper do not necessarily represent the views of the US DOE or the United States Government. Part of this work was performed at the Stanford Nano Shared Facilities (SNSF), supported by the National Science Foundation under award ECCS-2026822. Sandia National Laboratories is operated for the US DOE's National Nuclear Security Administration under contract DE-NA0003525. This research used resources of the Advanced Light Source, which is a DOE Office of Science User Facility under contract no. DE-AC02-05CH11231. S.R. was supported by the Air Force Office of Scientific Research (Grant FA9550-23-1-0215) for the growth of sputtered LCO films.

## Author contributions

K.S.W. designed the study concept. K.S.W., A.Z., T.D.B., and A.A.T. performed electrical measurements. A.Z., E.J.F., R.S.B., and S.R. fabricated the device. A.A., X.Q., and R.A. performed phase field modeling. K.S.W., M.P., and S.K. performed simulations. L.T. oversaw the design, assembly, and operation of the hardware for the Hopfield network. K.S.W. and S.K. performed Hopfield network experiments and related modeling. R.S.W. and S.K. supervised the entire project. K.S.W., R.S.W., and S.K. wrote the manuscript, and all authors commented on the manuscript.

## Competing interests

The authors declare no competing interests.
