## [Peer Review File · Nature Communications]

REVIEWER COMMENTS

Reviewer #1 (Remarks to the Author):

In this manuscript, K. S. Woo et al. present an LCO-based TRNG and its practical application to solving combinatorial optimization problems. The authors have developed a new type of TRNG device and achieved top performance in terms of speed. In this regard, I believe that this study will have a sufficient impact on the field. However, for the paper to be published, the following questions must be addressed:

My main points concerning the manuscript are as follows:

1. Although the authors explained and observed the thermal runaway characteristic in Fig. 1, the correlation between spin crossover and joule heating is still unclear. As far as I know, spin crossover typically does not result in a decrease in resistance. Is the NDR phenomenon observed here attributed to spin crossover? Alternatively, is NDR caused by thermal runaway? If NDR is solely due to thermal runaway, what function does spin crossover serve?
2. Did the author use an external capacitor? If not, how much is the device capacitance? Is it sufficient to drive the oscillation?
3. In the benchmarking table, please include a comparison of other performance metrics such as energy consumption, endurance, and other relevant factors. In particular, it is imperative to discuss the energy consumption, as this TRNG operates at high voltages and currents.
4. On page 6, the authors wrote that “Kim et al. similarly leveraged the self-clocking ability of a NbO₂ memristor⁵. Their approach, however, required a clock signal that was applied directly to the memristor instead of using a constant bias”. This misleads as if there is no external clock signal used in the LTO-based TRNG. In reality, both TRNGs utilize the same external clock signal but in different positions.
5. In practical applications, the proposed LCO-based TRNG may encounter speed issues compared to software-generated PRNGs, which operate faster from MHz to GHz. Therefore, comparing network performance solely based on the pre-generated bitstream without considering generation time may not be meaningful. Do the authors have any opinions or solutions regarding this issue?
6. The authors described the performance benefits of true random noise within simulated annealing, specifically local minimum escape. Is this a broadly applicable concept, or does it specifically pertain to

the maximum cut problem? To substantiate this description, please provide relevant existing studies or clarify its scope.

7. Please clarify the descriptions of Figure 4e and 4g in the text or captions for better understanding.

Reviewer #2 (Remarks to the Author):

In this work, the authors create stochastic oscillations in an LCO device, based on a DC current applied to the device. The stochasticity in the oscillation phase is attributed to a relatively slow transition between a low and high spin state vs. temperature, with energy similar to room temperature, causing instabilities. The random phase is used to randomly clock a flip flop to produce RNGs, and the resulting RNGs properties such as randomness are studied, as well as put into a memristor crossbar array to aid in simulated annealing.

The paper is well written and results well presented. The paper combines application and physical understanding in a way that will interest the readers of the journal. The use of a slowly-transitioning spin crossover point as an RNG is novel. But, the authors should address the following points.

- The authors put all TRNGs in the context of phase transitions: “Research into TRNGs has attracted increased attention, with several phase transitions being employed for this purpose, such as Mott transitions, magnetic switching, etc.” Not all TRNGs are phase transitions, e.g. the magnetic switching example. The phase transition method should be benchmarked compared to other major methods of TRNGs, such as stochastic magnetic tunnel junction and spin transfer torque-based RNG magnetic tunnel junctions. The 50 kb/s should be benchmarked against these other types that are not phase transition based.
- Why were different sputter-deposited films chosen for XRD vs. PLD for the devices? 70 nm of PLD material should be thick enough for XRD observation.
- Associated with Fig. 1e, why would more gradual change in the spin state lead to Joule heating? The connection to Joule heating is unclear, since usually that depends on the current applied, but not on the spin state of the material.
- Endurance was reported as 12000 s without degradation. Endurance should be reported in number of cycles, not seconds. It is hard to say if this is good endurance e.g. compared to nonvolatile memories. What will limit the endurance out to cycles of closer comparison such as 10^9 - 10^{15} cycles?

- Can Table 1 include other benchmarking parameters such as energy, delay, and endurance?
- Can the discussion of the physical origin of the effect (Fig. 3) be more closely connected to understanding of potential endurance, sensitivity of the TRNG to room temperature fluctuations, scaling to small sizes, and limits of energy and delay time? All of these properties do not have to be solved right now, but it is about understanding the path of the device if future research is done.
- With Fig. 4, please more explicitly explain the role of the LCO noise in the results. I expected to see something like Fig. 4i with and without LCO noise added to the simulated annealing experiment. In this part, the role of the LCO in achieving the function desired is left unclear.
- Could the RNG fluctuations be tuned? I.e. so the flip-flop spends 60% of time in one state vs. the other? Tunable TRNGs are particularly beneficial since rejection sampling to tune PRNGs is very time and energy intensive.
- It could be because the images were being viewed on a Mac, but grey lines are seen around some of the figures in Fig. 1.

Reviewer #3 (Remarks to the Author):

In the work “True Random Number Generation using the Spin Crossover in LaCoO₃”, the authors show how the spin crossover effect in LCO leads to a region of negative differential resistance, in which the system undergoes stochastic oscillations, which the authors propose for true random number generation.

The TRNG proposed is similar to that of (Kim et al , Nature Communications volume 12, Article number: 2906 (2021)) with the authors arguing that the gradual region of NDR being advantageous over a 1st order Mott transition for reasons of device endurance, and larger output. The authors discuss the origin of the effect via simulations and present it in the context of a Hopfield network.

The work is well written and clearly presented, and covers a broad range of topics from materials, to electrical characterisation and circuit design. I would recommend publication after revision.

In general, the work should be applauded for covering a broad range of research areas, however, I feel that leads to a lack of clarity in parts.

For example, the phrase

“To statistically quantify the variations, we measured the time period of 800 oscillations from 4 LCO components, revealing a substantial variation of roughly 25% (Fig. 2c).”

The authors measured 4 different devices 800 times and the variation is a matter of device to device variation? I don't believe this is what they mean, but that the variation is due to the stochasticity, but I find the phrasing vague, and more effort (for example in the supplementary) should be made to clarify this point. In general the supplementary data is presented with limited text, and perhaps could be built upon.

I did not get a clear idea if this report is consistent across a range of devices with potentially pronounced variation between devices or the report of a single device. The authors should comment on this.

I am not sure why the authors included the FFT in figure 2b, they do not seem to comment on it. I was unclear as to what was the timescales in the system. The stochastic oscillator has a frequency above 1MHz but the bit rate was 50Kbs-1. What was the limit to the bit rate? In supplementary 3 they show that the probability does not stay above a threshold for lower times, but I do not see any explanation of this effect. Is this due to the circuit design?

I feel that there is also insufficient description of the implementation of the noise into the Hopfield network, this should be expanded in the text or supplementary. Also the justification of the noise is that it is better than the Hopfield network in the absence of noise, but they do not show a comparison of this in the paper to see exactly the benefit of the noise, as the authors clearly state that the chip itself has already been demonstrated elsewhere.

I also don't really understand why the authors choose to plot Δ success probability for the comparison with the PRNG, and to be honest I find Fig 4j quite difficult to get anything significant out of. Perhaps integrating the figures 8/9 from the supplementary info, and including the noise level = 0 as well would be a good option?

In addition some smaller less significant points

- The authors should put more effort into placing their results within the broader TRNG community.- How does 50kbs-1 compare? I believe there are several demonstrations of TRNGs in the Mbs-1. The authors should address this.

“Our TRNG requires only a single circuit component, besides the LCO memristor for binary bit generation and achieves the highest bit generation rate of 50 kb s-1 among reported volatile-memristor-based TRNGs^{3–5,10}.”

- The authors have several declarative statements which I think should be accompanied by suitable references.

“Research into TRNGs has attracted increased attention, with several phase transitions being employed for this purpose, such as Mott transitions, magnetic switching, etc.”

“Our comprehensive approach revealed that the stochastic behavior, unlike in other phase transitions materials, is directly influenced by thermal fluctuations, which in turn introduce variations in material properties such as electrical conductivity.”

These sort of sentences really should be accompanied by references

- Insulator-to-metal transition (IMT) should be defined in the text

The paragraph of text “Memristors are increasingly employed as key components in TRNGs due to their inherent variabilities.” Appears after a lot of discussion of results, and I think would be more appropriate earlier in the manuscript.

March 31, 2024

Manuscript ID: NCOMMS-24-07799

Dear Reviewers,

We appreciate the valuable comments from the reviewers, which contributed significantly to enhancing the quality of the manuscript. We did our best to address all the comments and modified the manuscript accordingly. Our point-by-point responses to the comments are shown below. We hope the reviewers will find our revision a reasonable response to all the comments.

We welcome any additional comments that the reviewers may have.

Black (italics): Reviewer comments

Blue: Author responses

Red: Modified text in the manuscript/supplement.

Black (non-italic): original text in the manuscript/supplement.

Reviewer #1 (Remarks to the Author):

In this manuscript, K. S. Woo et al. present an LCO-based TRNG and its practical application to solving combinatorial optimization problems. The authors have developed a new type of TRNG device and achieved top performance in terms of speed. In this regard, I believe that this study will have a sufficient impact on the field. However, for the paper to be published, the following questions must be addressed:

My main points concerning the manuscript are as follows:

1. Although the authors explained and observed the thermal runaway characteristic in Fig. 1, the correlation between spin crossover and joule heating is still unclear. As far as I know, spin crossover typically does not result in a decrease in resistance. Is the NDR phenomenon observed here attributed to spin crossover? Alternatively, is NDR caused by thermal runaway? If NDR is solely due to thermal runaway, what function does spin crossover serve?

Answer: With increasing temperature, a spin crossover does lead to a decrease in resistance, as shown in both our measurements (Fig. 1d) and recent studies (Chiang, et al. *Low Temperature Physics* 46, 559 (2020), Galakhov, et al. *JETP Letters* 118, 189 (2023)). Further, there is a degree of nonlinearity associated with the decrease in resistance.

NDR requires two conditions: (1) increase in temperature upon increasing current (for thermally driven NDR) and (2) a minimum degree of nonlinearity in the resistance decreasing as a function of temperature. Our previous theoretical study established the exact degree of nonlinearity required and explained the conditions needed for NDR in detail (Brown et al., *Appl. Phys. Rev.* 9, 011308 (2022)). In this work, first, there is an increase in temperature upon increasing current due to Joule heating, which is expected. Second, there is a nonlinear decrease in resistance with increasing temperature due to a spin crossover. Because of a combination of both these processes, there is an NDR. We have expanded our discussion and provided additional citations to clarify the process.

Specific changes

- Clarification in Page 5 on the correlation between spin crossover and NDR.

(Added, main text) The gradual change in resistivity with temperature is also a signature of the spin-state transition (Fig. 1d)^{18,19}. The spin crossover process has a more gradual change in the resistance compared to an abrupt change in a first-order phase transition (e.g., in Mott insulators¹⁵). NDR requires two conditions – first, increase in temperature upon increasing current (for thermally driven NDR); second, a minimum degree of nonlinearity in the resistance decreasing as a function of temperature. Via *in-situ* thermal mapping at different current levels, we observed a relatively gradual temperature increase within the NDR region (Fig. 1e) in the order of ~20 K, satisfying the first criterion for NDR. Further, the three orders of magnitude decrease in resistance with increasing temperature (Fig. 1d), though gradual, provided sufficient nonlinearity to satisfy the second criterion required for NDR. Thus, the spin crossover is fundamentally responsible for the nature of the NDR and the dynamics associated with the NDR.

2. Did the author use an external capacitor? If not, how much is the device capacitance? Is it sufficient to drive the oscillation?

Answer: We did not use an external capacitor, and the oscillations were based on the device’s internal capacitance. We estimate the internal capacitance from the load resistance (2 k Ω) and the characteristic timescale of the oscillations of $\sim 0.5 \mu\text{s}$ (roughly proportional to the product of the resistance and the internal capacitance). With this information, we estimate the capacitance to be at most 0.3 nF.

Specific changes

- Inclusion of additional clarifying sentences in page 6.

(Added, main text) Since the oscillating time period ($\sim 0.5 \mu\text{s}$) is roughly equal to the product of the load resistance (2 k Ω) and the internal capacitor, we estimate the internal capacitor to be a maximum of 0.3 nF.

3. *In the benchmarking table, please include a comparison of other performance metrics such as energy consumption, endurance, and other relevant factors. In particular, it is imperative to discuss the energy consumption, as this TRNG operates at high voltages and currents.*

Answer: We agree with the reviewer that the comparison was not detailed enough. We added several items, such as device switching mechanism, TRNG endurance (number of bits produced per device), and energy consumption. We hope the advancement over prior approaches will now be apparent to the reviewer.

Specific changes

- Added more items in Table 1 for more detailed comparison.

Table 1 | Comparison of volatile-memristor-based TRNGs that passed NIST randomness test without post-processing. A low-power clock generator (CDCI6214, Texas Instruments) was assumed in the calculations (150 mW).

	This work	Jiang et al. ⁵	Woo et al. ⁶	Woo et al. ¹⁴	Kim et al. ³
Device switching mechanism	Non-first-order phase transition	Diffusive	Electronic switching	Diffusive	First-order phase transition
Source of randomness	Oscillations	Delay time	Delay & relaxation times	Delay & relaxation times	Oscillations
Bit generation rate (kb s⁻¹)	50	6	6	32	40

TRNG circuit components (# of components)	T flip-flop only (1)	Comparator, AND gate, 2 T flip-flops (4)	2 AND gates, T flip-flop (3)	XNOR gate, XOR gate, 4 D flip-flops (6)	Op-amp, T flip-flop (2)
TRNG endurance (# of bits produced per device)	600 M	54 M	Not reported (Two memristors scheme)	48 M	24 M
Energy consumption of clock generator (nJ b ⁻¹)	0	2.5×10^4	2.5×10^4	4.7×10^3	0

- Detailed explanation on energy consumption (Page 7).

(Added, main text) The overall energy consumption of a TRNG primarily depends on the number of active components, with each component consuming milliwatts of power. The self-oscillation-based TRNGs offer energy advantages by eliminating the need for a clock generator. While the energy consumption of the LCO-based memristor is higher compared to other recently demonstrated memristive devices, we anticipate the energy budget to be dominated by the peripheral components, leading the LCO-based TRNG to offer an overall lower power budget, since it requires fewer peripheral circuit components (Table 1).

4. On page 6, the authors wrote that “Kim et al. similarly leveraged the self-clocking ability of a NbO₂ memristor⁵. Their approach, however, required a clock signal that was applied directly to the memristor instead of using a constant bias”. This misleads as if there is no external clock signal used in the LTO-based TRNG. In reality, both TRNGs utilize the same external clock signal but in different positions.

Answer: We agree with the reviewer that this claim may be misleading. Therefore, we removed the claim of clock signal requirement in NbO₂ memristor.

Specific changes

- Removed the claim of clock signal requirement in NbO₂ memristor in Page 7.

(Modified, main text) Kim et al. similarly leveraged the self-clocking ability of a NbO₂ memristor³. Their approach, however, required an amplifier to increase the inherently low-current oscillating signal.

5. In practical applications, the proposed LCO-based TRNG may encounter speed issues compared to software-generated PRNGs, which operate faster from MHz to GHz. Therefore, comparing network performance solely based on the pre-generated bitstream without considering generation time may not be meaningful. Do the authors have any opinions or solutions regarding this issue?

Answer: We appreciate the reviewer for this critical comment. The reason for comparing the performance based on the bitstream is to verify the quality of truly random numbers, which have the potential to perform better in probabilistic computing. Competing against a mature technology (fabricated at sub-10-nm nodes) in terms of speed is unfair since this work is at the level of laboratory experiments (above 10 μm device sizes). The speeds are expected to increase notably upon size scaling and such speeds do not represent the fundamental difference between a TRNG and PRNG. Here we wanted to emphasize the fundamental difference between deterministic PRNG and stochastic TRNG. Separately, we also presented ways to further improve the bit generation rate in the ‘Stochastic oscillations in LCO’ section, but we now moved this part to ‘Using TRNGs to solve optimization problems’ section.

Specific changes

- We added these arguments and presented ways to increase the bit generation rate in Page 15.

(Added, main text) The fundamental distinction between deterministic PRNGs and stochastic TRNGs (in the quality of the random bit streams) highlights that TRNGs have superior performance in probabilistic computing. The speed of our TRNG (sub-MHz range) is far lower compared to prevailing CMOS technologies (up to GHz range). This difference is attributed mainly to the micrometer-scale sizes of our laboratory-scale devices compared to the CMOS technologies often manufactured at sub-10-nm sizes. As such, we expect the speed to increase notably upon scaling down the sizes of our prototype devices and not pose a fundamental bottleneck. Combining a feedback shift register or utilizing a nanoscale heater could further increase the bit generation rate^{28,44}.

6. The authors described the performance benefits of true random noise within simulated annealing, specifically local minimum escape. Is this a broadly applicable concept, or does it specifically pertain to the maximum cut problem? To substantiate this description, please provide relevant existing studies or clarify its scope.

Answer: The max-cut problem represents all possible optimization and constrained optimization problems, since it has full generality in terms of its representative Hamiltonian formulation (Lucas, *Frontiers in Physics*, 2, 5 (2014)). Thus, our ability to improve solutions to the max-cut problem is a demonstration of improving solutions to any optimization problem.

Specific changes

- We added these statements in Page 13.

7. Please clarify the descriptions of Figure 4e and 4g in the text or captions for better understanding.

Answer: We provided brief clarifications for Figs. 4e and 4g in their captions.

Specific changes

- Clarifications of experimental conductance-weight matrix in Fig. 4.

Reviewer #2 (Remarks to the Author):

In this work, the authors create stochastic oscillations in an LCO device, based on a DC current applied to the device. The stochasticity in the oscillation phase is attributed to a relatively slow transition between a low and high spin state vs. temperature, with energy similar to room temperature, causing instabilities. The random phase is used to randomly clock a flip flop to produce RNGs, and the resulting RNGs properties such as randomness are studied, as well as put into a memristor crossbar array to aid in simulated annealing.

The paper is well written and results well presented. The paper combines application and physical understanding in a way that will interest the readers of the journal. The use of a slowly-transitioning spin crossover point as an RNG is novel. But, the authors should address the following points.

- The authors put all TRNGs in the context of phase transitions: “Research into TRNGs has attracted increased attention, with several phase transitions being employed for this purpose, such as Mott transitions, magnetic switching, etc.” Not all TRNGs are phase transitions, e.g. the magnetic switching example. The phase transition method should be benchmarked compared to other major methods of TRNGs, such as stochastic magnetic tunnel junction and spin transfer torque-based RNG magnetic tunnel junctions. The 50 kb/s should be benchmarked against these other types that are not phase transition based.*

Answer: Thank you for these valuable comments. We agree that magnetic switching is not based on phase transition, so we modified the introductory part.

Regarding the TRNG comparison, in the revision, we considered different TRNGs, which can be differentiated by not only TRNG generation methods but also switching mechanisms of memristors (Diffusive, electron-based, first-order phase transition, and non-first-order phase transition). To make this point clear, we added device switching mechanism in the new comparison table (Table 1). The reason we did not include non-volatile memristors, such as magnetic tunnel junction is that the research focus has been shifted to volatile-memristor-based TRNGs since nonvolatile memory requires RESET process. Further, a lot of reported TRNGs have not been able to pass the NIST randomness test without post-processing. Therefore, we focused on different volatile-memristor-based TRNGs that passed the NIST

tests with no post-processing step. We added the following comments to make this point clear.

Specific changes

- Modified introductory part in Page 2.

(Modified, main text) Research into TRNGs has attracted increased attention, with several switching mechanisms being employed for this purpose, such as Mott transitions³, magnetic switching⁴, etc.

- Additional explanation for comparing volatile-memristor-based TRNGs (Page 8).

(Added, main text) However, these TRNG approaches face practical challenges, including circuit complexity, requirement of RESET process and reliance on post-processing steps, creating challenges for on-chip integration. To address these issues, there has been a shift in focus towards volatile-memristor-based TRNGs with self-OFF switching behavior, which can reduce the energy consumption. Therefore, we compare the performance of volatile-memristor-based TRNGs that passed the NIST randomness test without post-processing (Table 1).

- Added device switching mechanism in Table 1.

• *Why were different sputter-deposited films chosen for XRD vs. PLD for the devices? 70 nm of PLD material should be thick enough for XRD observation.*

Answer: We performed scanning transmission X-ray microscopy (STXM) in the oxygen K edge, which is a transmission technique. This technique requires films grown on a substrate that is transparent to x-rays in the oxygen K-edge energies (500-600 eV), such as suspended SiN membranes. PLD films cannot be grown on such substrates, which is why we used sputtering for our x-ray measurements.

Specific changes

- Added the above text to Fig. 1c caption.

• *Associated with Fig. 1e, why would more gradual change in the spin state lead to Joule heating? The connection to Joule heating is unclear, since usually that depends on the current applied, but not on the spin state of the material.*

Answer: We agree that the connection to Joule heating was unclear in the original version of the manuscript. As the reviewer mentioned, the applied bias can indeed lead to Joule heating, which in turn heats up the device and causes spin crossover.

We made corrections regarding the correlation between spin crossover and NDR to prevent confusion.

Specific changes

- Clarification in Page 5 on the spin crossover and NDR.

(Added, main text) The gradual change in resistivity with temperature is also a signature of the spin-state transition (Fig. 1d)^{18,19}. The spin crossover process has a more gradual change in the resistance compared to an abrupt change in a first-order phase transition (e.g., in Mott insulators¹³). NDR requires two conditions – first, increase in temperature upon increasing current (for thermally driven NDR); second, a minimum degree of nonlinearity in the resistance decreasing as a function of temperature. Via *in-situ* thermal mapping at different current levels, we observed a relatively gradual temperature increase within the NDR region (Fig. 1e) in the order of ~20 K, satisfying the first criterion for NDR. Further, the three orders of magnitude decrease in resistance with increasing temperature (Fig. 1d), though gradual, provided sufficient nonlinearity to satisfy the second criterion required for NDR. Thus, the spin crossover is fundamentally responsible for the nature of the NDR and the dynamics associated with the NDR.

• *Endurance was reported as 12000 s without degradation. Endurance should be reported in number of cycles, not seconds. It is hard to say if this is good endurance e.g. compared to nonvolatile memories. What will limit the endurance out to cycles of closer comparison such as 10⁹-10¹⁵ cycles?*

Answer: As the reviewer pointed out, endurance is generally verified by continuously switching the device into ON and OFF states with applied pulses, and most of the reported memristor-based TRNGs employ device switching behaviors as the random sources. Therefore, it is reasonable to evaluate the endurance in the conventional way. In this case, however, we are dealing with self-oscillating property of memristor by applying constant bias. For endurance testing, we measured oscillating time, which can be converted into total number of bits that can be generated, which in our case was 600 M bits. Instead of comparing device endurance, we included the number of bits produced per device, termed TRNG endurance, in the comparison table.

Specific changes

- Added TRNG endurance in Table 1.
- Modified texts in Page 7.

(Modified, main text) Furthermore, our TRNG exhibits good endurance in that the LCO component oscillated over 12,000 seconds without any degradation, proving its capability to generate at least 600 M bits (Supplementary Fig. 4).

• *Can Table 1 include other benchmarking parameters such as energy, delay, and endurance?*

Answer: We agree that the comparison was not detailed enough. We included other items, such as device switching mechanism, TRNG endurance (number of bits produced per device), and energy consumption. We hope the advancement over prior approaches will now be apparent to the reviewer.

Specific changes

- More detailed comparison in Table 1.

Table 1 | Comparison of volatile-memristor-based TRNGs that passed NIST randomness test without post-processing. A low-power clock generator (CDCI6214, Texas Instruments) was assumed in the calculations (150 mW).

	This work	Jiang et al. ⁵	Woo et al. ⁶	Woo et al. ¹⁴	Kim et al. ³
--	------------------	---------------------------	-------------------------	--------------------------	-------------------------

Device switching mechanism	Non-first-order phase transition	Diffusive	Electronic switching	Diffusive	First-order phase transition
Source of randomness	Oscillations	Delay time	Delay & relaxation times	Delay & relaxation times	Oscillations
Bit generation rate (kb s⁻¹)	50	6	6	32	40
TRNG circuit components (# of components)	T flip-flop only (1)	Comparator, AND gate, 2 T flip-flops (4)	2 AND gates, T flip-flop (3)	XNOR gate, XOR gate, 4 D flip-flops (6)	Op-amp, T flip-flop (2)
TRNG endurance (# of bits produced per device)	600 M	54 M	Not reported (Two memristors scheme)	48 M	24 M
Energy consumption of clock generator (nJ b⁻¹)	0	2.5×10^4	2.5×10^4	4.7×10^3	0

- Detailed explanation on energy consumption (Page 6).

(Added, main text) The overall energy consumption of a TRNG primarily depends on the number of active components, with each component consuming milliwatts of power. The self-oscillation-based TRNGs offer energy advantages by eliminating the need for a clock generator. While the energy consumption of the LCO-based memristor is higher compared to other recently demonstrated memristive devices, we anticipate the energy budget to be dominated by the peripheral components, leading the LCO-based TRNG to offer an overall lower power budget, since it requires fewer peripheral circuit components (Table 1).

• *Can the discussion of the physical origin of the effect (Fig. 3) be more closely connected to understanding of potential endurance, sensitivity of the TRNG to room temperature fluctuations, scaling to small sizes, and limits of energy and delay time? All of these properties do not have to be solved right now, but it is about understanding the path of the device if future research is done.*

Answer: We added more discussion on Fig. 3. We have also referenced our past theoretical work, which demonstrates that an abrupt switching leads to lower endurance (due to damages, thermal runaways, etc.). In addition, we have also commented on the transition temperature of LCO being favorable, unlike the usually used Mott insulators, which have transition temperatures that are either too high or too low.

Specific changes

- Additional explanation in page 11 on Fig. 3, and an additional reference.

(Added, main text) This large variation is the key factor that contributes to the stochastic oscillations even at room temperature. Furthermore, there is no sudden change in high spin fraction at any specific temperature, unlike first-order phase transition materials, which have abrupt transitions causing structural damages during the switching¹⁶. In addition, Mott insulators that are routinely used to build oscillators undergo a transition at either very high temperatures (above 1000 K in the case of NbO_2 ¹²) or very low temperatures (about 340 K in VO_2 ¹³). Such transition temperatures are below the standard operating ambient temperature for commercial electronics (about 350 K) or very high (potentially damaging nearby materials if switching temperature is above 1000 K). LCO, on the other hand, has a transition in a broad range from room temperature up to about 700 K, which makes it suitable for chip operating environments. Therefore, LCO is a more stable on-chip material, as verified by our endurance testing and owing to its favorable transition temperature.

• *With Fig. 4, please more explicitly explain the role of the LCO noise in the results. I expected to see something like Fig. 4i with and without LCO noise added to the simulated annealing experiment. In this part, the role of the LCO in achieving the function desired is left unclear.*

Answer: Thank you for this comment. We agree that demonstration of Hopfield network without noise should be presented for better understanding its role. Without the noise, the success probability will just be zero as the network converges to local minima or a wrong solution (instead of a global minimum, or the correct solution). Therefore, instead of adding data in Fig. 4i, we added energy descent of the network without any noise in Fig. 4h.

Specific changes

- Additional data for no noise in Fig. 4h.
- Added explanation on the data for no noise (Page 15).
- Added additional context in the figure caption of Fig. 4.

(Added, main text) Without any noise, the Hopfield network converged to a local minimum after a few cycles and could not escape from this state. Therefore, the noise is indispensable in solving NP-hard problems that have complex energy landscapes.

- *Could the RNG fluctuations be tuned? I.e. so the flip-flop spends 60% of time in one state vs. the other? Tunable TRNGs are particularly beneficial since rejection sampling to tune PRNGs is very time and energy intensive.*

Answer: We appreciate the reviewer for this valuable comment. Since the generated bit is based on the number of oscillations, the RNG fluctuations can be tuned by reducing the oscillating bias or toggle (T) input pulse time, as demonstrated in Supplementary Figs 1 and 3. We added this statement to point out this advantage.

Specific changes

- Additional notes on the tunability of this TRNG design in Page 7.

(Added, main text) Moreover, since the generated bit is based on the number of oscillations (bit flipping), the randomness of our TRNG can be tuned by adjusting the oscillating bias or T input pulse time (Supplementary Figs 1 and 3). This tunable TRNG may present an efficient alternative to the time-consuming and energy-intensive process of rejection sampling used with PRNGs.

• *It could be because the images were being viewed on a Mac, but grey lines are seen around some of the figures in Fig. 1.*

Answer: Thank you for pointing to this issue. We will work with the editorial office to ensure correct formatting.

Reviewer #3 (Remarks to the Author):

In the work “True Random Number Generation using the Spin Crossover in LaCoO₃”, the authors show how the spin crossover effect in LCO leads to a region of negative differential resistance, in which the system undergoes stochastic oscillations, which the authors propose for true random number generation.

The TRNG proposed is similar to that of (Kim et al , Nature Communications volume 12, Article number: 2906 (2021)) with the authors arguing that the gradual region of NDR being advantageous over a 1st order Mott transition for reasons of device endurance, and larger output. The authors discuss the origin of the effect via simulations and present it in the context of a Hopfield network.

The work is well written and clearly presented, and covers a broad range of topics from materials, to electrical characterisation and circuit design. I would recommend publication after revision.

In general, the work should be applauded for covering a broad range of research areas, however, I feel that leads to a lack of clarity in parts.

For example, the phrase

“To statistically quantify the variations, we measured the time period of 800 oscillations from 4 LCO components, revealing a substantial variation of roughly 25% (Fig. 2c).”

The authors measured 4 different devices 800 times and the variation is a matter of device to device variation? I don't believe this is what they mean, but that the variation is due to the stochasticity, but I find the phrasing vague, and more effort (for example in the supplementary) should be made to clarify this point. In general the supplementary data is presented with limited text, and perhaps could be built upon.

I did not get a clear idea if this report is consistent across a range of devices with potentially pronounced variation between devices or the report of a single device. The authors should comment on this.

Answer: We appreciate the reviewer for this careful comment. We measured 4 different devices to show similar stochastic variations in multiple devices, ensuring that the observed phenomena are not limited to a single device. We added an additional description to clarify this measurement.

Specific changes

- Additional description on Fig. 2c (Page 6).

(Added, main text) To statistically quantify the variations, we measured the time period of 800 oscillations from a single LCO component, revealing a substantial variation of roughly 25% (from the central time period) within a given component (Fig. 2c). We repeated this measurement on four different devices, and all measured devices exhibited similar stochastic variations, ensuring that the observed phenomena are not limited to a single device.

I am not sure why the authors included the FFT in figure 2b, they do not seem to comment on it. I was unclear as to what was the timescales in the system. The stochastic oscillator has a frequency above 1MHz but the bit rate was 50Kbs-1. What was the limit to the bit rate? In supplementary 3 they show that the probability does not stay above a threshold for lower times, but I do not see any explanation of this effect. Is this due to the circuit design?

Answer: Thank you for the detailed comment. We included the FFT data to approximate the oscillation frequency (~2 MHz). As the reviewer pointed out, the bit generation rate is slower than the oscillation frequency. This is because the bit generation is based on the multiple oscillations produced by the memristor. These oscillations are stochastic and therefore generate random bits. The bit generation rate can be increased by reducing toggle (T) input pulse time (thereby approaching a bit generation rate equal to that of the oscillating frequency). However, reducing the input pulse time means that there will be lesser number of oscillations (bit flipping) before producing the output bit. When the input pulse time was reduced to 5 μ s, the resulting randomness significantly decreased and failed to pass the NIST randomness test at below 5 μ s, as shown in Supplementary Fig. 3. However, there may be applications where data streams with limited randomness may be useful, wherein such a reduced T input pulse time would offer faster bit rates. We have mentioned this aspect in the revision.

Specific changes

- Additional description on Fig. 2b caption and Supplementary Fig. 3.

(Added, main text) **Fig. 2** | LCO-based TRNG. **a**, Twenty sequential oscillations at $I_{ext} = 3.2$ mA. **b**, Fourier transform of the first oscillation in **a** to approximate the oscillation frequency (~2 MHz).

(Added, supplement) **Supplementary Fig. 3** | **Frequency (monobit) test results at different toggle (T) input pulse times.** The bit generation rate can be increased by reducing T input pulse time, but there will be lesser number of oscillations (bit flipping) before producing the output bit. When the time was reduced to 5 μ s, the resulting randomness significantly decreased and failed to pass the test at below 5 μ s. However, there may be applications where data streams with limited randomness may be useful, wherein such a reduced T input pulse time would offer faster bit rates.

I feel that there is also insufficient description of the implementation of the noise into the Hopfield network, this should be expanded in the text or supplementary. Also the justification of the noise is that it is better than the Hopfield network in the absence of noise, but they do not show a comparison of this in the paper to see exactly the benefit of the noise, as the authors clearly state that the chip itself has already been demonstrated elsewhere.

Answer: We agree that demonstration of Hopfield network without noise should be presented for better understanding its role. Without the noise, the success probability will just be zero as the network converges to local minima. Therefore, we added energy descent of the network without any noise in Fig. 4h.

Specific changes

- Additional data for no noise in Fig. 4h.
- Added explanation on the data for no noise (Page 15).

(Added, main text) Without any noise, the Hopfield network converged to a local minimum after a few cycles and could not escape from this state. Therefore, the noise is indispensable in solving NP-hard problems that have complex energy landscapes.

I also don't really understand why the authors choose to plot Δ success probability for the comparison with the PRNG, and to be honest I find Fig 4j quite difficult to get anything significant out of. Perhaps integrating the figures 8/9 from the supplementary info, and including the noise level = 0 as well would be a good option?

Answer: We believe Fig. 4j brings out significant results demonstrating that TRNG can outperform PRNG in most cases. The reason for comparing the performance between them is to verify the quality of truly random numbers, which have the potential to perform better in probabilistic computing. We wanted to emphasize the fundamental difference between deterministic PRNG and stochastic TRNG. Supplementary Figs. 8 and 9 show the performance at 6000 and 18000 iterations, respectively, so they were separated into two figures. As addressed in the previous comment, we added data on energy descent when the noise level is 0 (Fig. 4h).

Specific changes

- Modification of Fig. 4j to illustrate the interpretation of the plot.
- Additional clarification in the caption of Fig. 4j.
- Additional description on Fig. 4j (Page 15).

j, Success probability of TRNG-based network minus that of PRNG-based network at different node sizes. Data points above zero on the vertical axis indicate superior performance compared to PRNGs.

(Added, main text) The fundamental distinction between deterministic PRNGs and stochastic TRNGs (in the quality of the random bit streams) highlights that TRNGs have superior performance in probabilistic computing. The speed of our TRNG (sub-MHz range) is far lower compared to prevailing CMOS technologies (up to GHz range). This difference is attributed mainly to the micrometer-scale sizes of our laboratory-scale devices compared to the CMOS technologies often manufactured at sub-10-nm sizes. As such, we expect the speed to increase notably upon scaling down the sizes of our prototype devices and not pose a fundamental bottleneck.

In addition some smaller less significant points

- *The authors should put more effort into placing their results within the broader TRNG community.- How does 50kbs-1 compare? I believe there are several demonstrations of TRNGs in the Mbs-1. The authors should address this.*

“Our TRNG requires only a single circuit component, besides the LCO memristor for binary bit generation and achieves the highest bit generation rate of 50 kb s-1 among reported volatile-memristor-based TRNGs3–5,10.”

Answer: The reason we did not include nonvolatile memories from our study is that the research focus has been shifted to volatile-memristor-based TRNGs since nonvolatile-memory necessitates RESET process. TRNGs in the Mbs⁻¹ usually requires complex circuit (large number of components) with multiple clock signals, which are not suitable to be in this comparison. However, we considered different volatile-memristor-based TRNGs, which can be differentiated by not only TRNG generation methods but also switching mechanisms of memristors (Diffusive, electron-based, first-order phase transition, and non-first-order phase transition). Since a lot of reported TRNGs have not been able to pass the NIST randomness test without post-processing, we focused on different volatile-memristor-based TRNGs that passed the NIST tests without post-processing. However, we agree with the reviewer’s suggestion to put more details in our comparison. We included several items (switching mechanism, TRNG endurance, energy consumption) in Table 1 for more detailed comparison.

Specific changes

- Additional explanation for comparing volatile-memristor-based TRNGs (Page 8).

(Added, main text) However, these TRNG approaches face practical challenges, including circuit complexity, requirement of RESET process and reliance on post-processing steps, creating challenges for on-chip integration. To address these issues, there has been a shift in focus towards volatile-memristor-based TRNGs with self-OFF switching behavior, which can reduce the energy consumption. Therefore, we compare the performance of volatile-memristor-based TRNGs that passed the NIST randomness test without post-processing (Table 1).

- More detailed comparison in Table 1.

Table 1 | Comparison of volatile-memristor-based TRNGs that passed NIST randomness test without post-processing. A low-power clock generator (CDCI6214, Texas Instruments) was assumed in the calculations (150 mW).

	This work	Jiang et al. ⁵	Woo et al. ⁶	Woo et al. ¹⁴	Kim et al. ³
Device switching mechanism	Non-first-order phase transition	Diffusive	Electronic switching	Diffusive	First-order phase transition

Source of randomness	Oscillations	Delay time	Delay & relaxation times	Delay & relaxation times	Oscillations
Bit generation rate (kb s ⁻¹)	50	6	6	32	40
TRNG circuit components (# of components)	T flip-flop only (1)	Comparator, AND gate, 2 T flip-flops (4)	2 AND gates, T flip-flop (3)	XNOR gate, XOR gate, 4 D flip-flops (6)	Op-amp, T flip-flop (2)
TRNG endurance (# of bits produced per device)	600 M	54 M	Not reported (Two memristors scheme)	48 M	24 M
Energy consumption of clock generator (nJ b ⁻¹)	0	2.5×10^4	2.5×10^4	4.7×10^3	0

- Detailed explanation on energy consumption (Page 7).

(Added, main text) The overall energy consumption of a TRNG primarily depends on the number of active components, with each component consuming milliwatts of power. The self-oscillation-based TRNGs offer energy advantages by eliminating the need for a clock generator. While the energy consumption of the LCO-based memristor is higher compared to other recently demonstrated memristive devices, we anticipate the energy budget to be dominated by the peripheral components, leading the LCO-based TRNG to offer an overall lower power budget, since it requires fewer peripheral circuit components (Table 1).

• *The authors have several declarative statements which I think should be accompanied by suitable references.*

“Research into TRNGs has attracted increased attention, with several phase transitions being employed for this purpose, such as Mott transitions, magnetic switching, etc.”

“Our comprehensive approach revealed that the stochastic behavior, unlike in other phase transitions materials, is directly influenced by thermal fluctuations, which in turn introduce variations in material properties such as electrical conductivity.”

These sort of sentences really should be accompanied by references

Answer: We included references following the reviewer’s suggestion. We also included additional references in several other places in the manuscript.

- *Insulator-to-metal transition (IMT) should be defined in the text*

The paragraph of text “Memristors are increasingly employed as key components in TRNGs due to their inherent variabilities.” Appears after a lot of discussion of results, and I think would be more appropriate earlier in the manuscript.

Answer: Thank you for the feedback.

Specific changes

- Insulator-to-metal transition (IMT) is defined (Page 2).
- Moved the suggested paragraph to ‘Stochastic oscillations in LCO’ section (Page 8).

REVIEWER COMMENTS

Reviewer #1 (Remarks to the Author):

I am satisfied with the authors' response. I think this paper is suitable for publication.

Reviewer #2 (Remarks to the Author):

The authors have thoroughly addressed most of my comments. I do have a few remaining points that I think should be addressed before publication. Below I copied my original comments, the authors' response, and the new, remaining comments at the bottom.

Reviewer #2:

- The authors put all TRNGs in the context of phase transitions: "Research into TRNGs has attracted increased attention, with several phase transitions being employed for this purpose, such as Mott transitions, magnetic switching, etc." Not all TRNGs are phase transitions, e.g. the magnetic switching example. The phase transition method should be benchmarked compared to other major methods of TRNGs, such as stochastic magnetic tunnel junction and spin transfer torque-based RNG magnetic tunnel junctions. The 50 kb/s should be benchmarked against these other types that are not phase transition based.

Answer: Thank you for these valuable comments. We agree that magnetic switching is not based on phase transition, so we modified the introductory part.

Regarding the TRNG comparison, in the revision, we considered different TRNGs, which can be differentiated by not only TRNG generation methods but also switching mechanisms of memristors (Diffusive, electron-based, first-order phase transition, and non-first-order phase transition). To make this point clear, we added device switching mechanism in the new comparison table (Table 1). The reason we did not include non-volatile memristors, such as

magnetic tunnel junction is that the research focus has been shifted to volatile-memristor-based TRNGs since nonvolatile memory requires RESET process. Further, a lot of reported TRNGs have not been able to pass the NIST randomness test without post-processing. Therefore, we focused on different volatile-memristor-based TRNGs that passed the NIST8 tests with no post-processing step. We added the following comments to make this point clear.

Specific changes

- Modified introductory part in Page 2.

(Modified, main text) Research into TRNGs has attracted increased attention, with several switching mechanisms being employed for this purpose, such as Mott transitions³, magnetic switching⁴, etc.

- Additional explanation for comparing volatile-memristor-based TRNGs (Page 8).

(Added, main text) However, these TRNG approaches face practical challenges, including circuit complexity, requirement of RESET process and reliance on post-processing steps, creating challenges for on-chip integration. To address these issues, there has been a shift in focus towards volatile-memristor-based TRNGs with self-OFF switching behavior, which can reduce the energy consumption. Therefore, we compare the performance of volatile-memristor-based TRNGs that passed the NIST randomness test without post-processing (Table 1).

- Added device switching mechanism in Table 1.

Remaining comments:

- I understand the point that many high-energy-barrier, nonvolatile solutions require a reset, and in this work a reset is avoided, which is important to emphasize. But, that does not mean the alternative solutions lose out on speed or that they do not pass the NIST tests, e.g. see this paper that uses a high-energy-barrier STT-MRAM and passes most of the NIST tests:

<https://journals.aps.org/prapplied/abstract/10.1103/PhysRevApplied.19.024035>. In that paper, they report 50 Mb/s, so it is not clear that needing a reset makes it not competitive. Can you further clarify

why you still do not compare to these types in the table? It seems the speed is orders of magnitude off, unless I am missing something.

- In Table 1, I do not see the device energy consumption in the comparison table, just the energy consumption of the clock generator.
- I think Table 1 is also missing comparison to MTJ-based p-bits that are using low energy barriers. E.g. this work also recently in Nature Comm: <https://www.nature.com/articles/s41467-024-46645-6>. You can find a number of similar papers to get speed and energy numbers (as well as NIST tests validation), at a brief glance at that one I see $\sim 10^4$ flips/ns, so I think these also need to be clearly compared against for bit generation rate and energy consumption. You need to make clear the benefits and drawbacks of the LCO approach against these two MTJ types (high energy barrier and low energy barrier) to make the novelty, benefit, and use-case-fit of using the spin state transition clear.

Reviewer #3 (Remarks to the Author):

The authors have addressed my concerns and i feel that the manuscript is ready for publication

Response to Referees

Black (italics): Reviewer comments

Blue: Author responses

Reviewer #1 (Remarks to the Author):

I am satisfied with the authors' response. I think this paper is suitable for publication.

Reviewer #3 (Remarks to the Author):

The authors have addressed my concerns and i feel that the manuscript is ready for publication.

Thank you for the positive feedback on our paper.

Reviewer #2 (Remarks to the Author):

The authors have thoroughly addressed most of my comments. I do have a few remaining points that I think should be addressed before publication. Below I copied my original comments, the authors' response, and the new, remaining comments at the bottom.

Reviewer #2:

- The authors put all TRNGs in the context of phase transitions: “Research into TRNGs has attracted increased attention, with several phase transitions being employed for this purpose, such as Mott transitions, magnetic switching, etc.” Not all TRNGs are phase transitions, e.g. the magnetic switching example. The phase transition method should be benchmarked compared to other major methods of TRNGs, such as stochastic magnetic tunnel junction and spin transfer torque-based RNG magnetic tunnel junctions. The 50 kb/s should be benchmarked against these other types that are not phase transition based.*

Answer: Thank you for these valuable comments. We agree that magnetic switching is not based on phase transition, so we modified the introductory part.

Regarding the TRNG comparison, in the revision, we considered different TRNGs, which can be differentiated by not only TRNG generation methods but also switching mechanisms of memristors (Diffusive, electron-based, first-order phase transition, and non-first-order phase transition). To make this point clear, we added device switching mechanism in the new comparison table (Table 1). The reason we did not include non-volatile memristors, such as magnetic tunnel junction is that the research focus has been shifted to volatile-memristor-based TRNGs since nonvolatile memory requires RESET process. Further, a lot of reported TRNGs have not been able to pass the NIST randomness test without post-processing. Therefore, we focused on different volatile-memristor-based TRNGs that passed the NIST8 tests with no post-processing step. We added the following comments to make this point clear.

Specific changes

- Modified introductory part in Page 2.*

(Modified, main text) Research into TRNGs has attracted increased attention, with several switching mechanisms being employed for this purpose, such as Mott transitions³, magnetic switching⁴, etc.

- Additional explanation for comparing volatile-memristor-based TRNGs (Page 8).*

(Added, main text) However, these TRNG approaches face practical challenges, including

circuit complexity, requirement of RESET process and reliance on post-processing steps, creating challenges for on-chip integration. To address these issues, there has been a shift in focus towards volatile-memristor-based TRNGs with self-OFF switching behavior, which can reduce the energy consumption. Therefore, we compare the performance of volatile-memristor-based TRNGs that passed the NIST randomness test without post-processing (Table 1).

- *Added device switching mechanism in Table 1.*

Remaining comments:

All three of the reviewer's comments relate to performance comparisons, especially to magnetic devices. We provide our point-by-point responses to each of the comments below. In response to all three comments, we have added a new section (on Page 17) titled "Comparing reported TRNGs". We hope that this addition provides a fair and balanced comparison, which we believe the reviewer will appreciate.

• *I understand the point that many high-energy-barrier, nonvolatile solutions require a reset, and in this work a reset is avoided, which is important to emphasize. But, that does not mean the alternative solutions lose out on speed or that they do not pass the NIST tests, e.g. see this paper that uses a high-energy-barrier STT-MRAM and passes most of the NIST tests: <https://journals.aps.org/prapplied/abstract/10.1103/PhysRevApplied.19.024035>. In that paper, they report 50 Mb/s, so it is not clear that needing a reset makes it not competitive. Can you further clarify why you still do not compare to these types in the table? It seems the speed is orders of magnitude off, unless I am missing something.*

Answer: We appreciate the reviewer's comments to include nonvolatile solutions, such as STT-MRAM. Avoiding a RESET process is preferable in terms of speed and energy consumption (as illustrated by Jiang et al. *Nat. Commun.* 8, 882 (2017)). However, the absence of RESET process was not the only factor that we considered in our comparison. Most importantly, we focused on approaches that passed all the NIST tests without any post-processing (which we have now clarified in Table 1's caption). After carefully reviewing the suggested paper on high-energy-barrier STT-MRAM, we found that the device itself did not pass all the NIST tests and required post-processing steps (such as XOR operations, the

experimental performance of which remains unclear). Notably, the frequency (monobit) test, which the STT-MRAM-based work failed to pass, is essential. Failing this test implies possible failure in many other tests (as shown by Kim et al. *Nat. Commun.* 12, 2906 (2021), Rukhin et al. *NIST Special Publication 800-22* (2010)). According to the NIST Special Publication, at least 55 sequences must be processed for statistically meaningful results, but the STT-MRAM work only performed the NIST tests once. Further, the NIST tests were not performed at the bit generation rate of 50 Mb/s. We mentioned in the main text and Supplementary Fig. 3 that our speed assessment takes into account whether the frequency test was passed. Without passing the frequency test, the LCO-based TRNG could also reach ~Mb/s (but such a claim may be misleading). Since there are many reports of random number generators that demonstrate passing the NIST tests only with post-processing steps, we limited our comparison to approaches that do not require any post-processing, verifying the true randomness of the device's intrinsic stochasticity. We acknowledge that many of the reported random number generators may indeed pass the NIST tests if evaluated fully. However, to make a fair comparison, we chose only those that met the above criterion.

Specific changes: we have included the above discussion in the new section “Comparing reported TRNGs” (Page 17).

• *In Table 1, I do not see the device energy consumption in the comparison table, just the energy consumption of the clock generator.*

Answer: As we mentioned in our previous revision, we did not include device energy consumption because the energy budget is predominantly influenced by peripheral components, each consuming large amounts of power. On the other hand, the energy consumption of devices listed in the comparison table ranges from nanowatts or femtowatts. Furthermore, some devices, including our LCO, were used as clock signals (with a low-power clock generator generally consuming ~150 mW). However, we understand that our comparison could either be misleading or could exclude other types of TRNGs (e.g., those based on MTJs). It is difficult and unfair to compare reported energy and power numbers from academic research (the issues of which we have outlined in our new section “Comparing reported TRNGs” (Page 17)). Instead, we have provided as fair and balanced

comparison as possible in this section, and have also outlined the limitations of reported performance numbers.

• *I think Table 1 is also missing comparison to MTJ-based p-bits that are using low energy barriers. E.g. this work also recently in Nature*

Comm: <https://www.nature.com/articles/s41467-024-46645-6>. You can find a number of similar papers to get speed and energy numbers (as well as NIST tests validation), at a brief glance at that one I see $\sim 10^4$ flips/ns, so I think these also need to be clearly compared against for bit generation rate and energy consumption. You need to make clear the benefits and drawbacks of the LCO approach against these two MTJ types (high energy barrier and low energy barrier) to make the novelty, benefit, and use-case-fit of using the spin state transition clear.

Answer: Thank you for suggesting a new paper, which was not published at the time of our first revision. The reported quantity of 10^4 flips/ns was based on projections at the 7 nm technology node, while their experimental demonstrations were very slow. Also, the paper did not demonstrate passing of all NIST tests (or did not meet the criterion as mentioned in the response for the first comment).

Specific changes: As noted before, we have included a balanced comparison of different physical processes, including magnetic switching in the new section “Comparing reported TRNGs” (Page 17).

REVIEWERS' COMMENTS

Reviewer #2 (Remarks to the Author):

I appreciate the authors' additional discussion comparing their results to reported TRNGs. Their emphasis on passing all the NIST tests without post-processing well highlights the reported method. I agree the paper is ready for publication.